# Advanced Label-Free Laser Scanning Microscopy and Its Biological Imaging Application

**Xue Wang** [1,2]**, Xinchao Lu** [1,*] **and Chengjun Huang** [1,2,*]

1. Health Electronics Center, Institute of Microelectronics of the Chinese Academy of Sciences, Beijing 100029, China; wangxue@ime.ac.cn.com
2. School of Microelectronics, University of Chinese Academy of Sciences, Beijing 100049, China
*  Correspondence: luxinchao@ime.ac.cn (X.L.); huangchengjun@ime.ac.cn (C.H.)

**Abstract:** By eliminating the photodamage and photobleaching induced by high intensity laser and fluorescent molecular, the label-free laser scanning microscopy shows powerful capability for imaging and dynamic tracing to biological tissues and cells. In this review, three types of label-free laser scanning microscopies: laser scanning coherent Raman scattering microscopy, second harmonic generation microscopy and scanning localized surface plasmon microscopy are discussed with their fundamentals, features and recent progress. The applications of label-free biological imaging of these laser scanning microscopies are also introduced. Finally, the performance of the microscopies is compared and the limitation and perspectives are summarized.

**Keywords:** laser scanning microscopy; label-free; coherent anti-Stokes Raman scattering; stimulated Raman scattering; second harmonic generation; surface plasmon polaritons

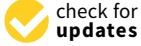



## 1. Introduction

To date, optical microscopy has been the most important tool in biology for observing the biological structures, morphology and dynamics in subcellular size. Due to high spatial resolution induced by focal point of light in point-scanning mode, laser scanning microscopy (LSM) got rapidly development in the past decades. In 1957, the first confocal laser scanning microscopy (CLSM) was proposed by M. Minsky for reducing blurring and improving contrast via point-to-point illumination and detection [1]. Nowadays, the fluorescent confocal laser scanning microscopy (FCLSM) that combines the fluorescent labels and CLSM has been one of the most powerful and versatile methods for studying cells, tissues and organisms. Furthermore, various super resolution microscopies have been developed based on FCLSM, such as stimulated emission depletion (STED) [2–4], reversible saturable optical fluorescence transitions (RESOLFT) [5–7], and saturated excitation (SAX) [8,9].

Although FCLSM has been applied extensively in biological imaging, both photodamage and photo-bleaching induced by high intensity laser and fluorophore hindered the imaging and analysis to biological samples. Furthermore, functional disturbance caused by fluorophore may also limit the applications of FCLSM. In contrast, advanced label-free imaging LSM techniques shows advantages in imaging and dynamic tracing to biological tissues and cells. This review discussed LSM and its label-free biological imaging application. Firstly, the laser scanning coherent Raman scattering microscopy (CRSM) and its two subclasses, i.e., coherent anti-Stokes Raman scattering microscopy (CARSM) and stimulated Raman scattering microscopy (SRSM) were introduced, and their applications in imaging cells and issues were reviewed and compared. Then, the progress of second harmonic generation microscopy (SHGM) was discussed. Finally, label-free scanning localized surface plasmon microscopy (SLSPM) based on LSM was introduced, which was especially suitable for studying morphology and dynamics at cell-subtract interface. These three LSMs based microscopic techniques were summarized and compared in the aspects

of work wavelength, lateral resolution, axial resolution, etc. The advantages and potential applications for biological imaging of each technique were discussed.

## 2. Laser Scanning Coherent Raman Scattering Microscopy (CRSM)

Due to the vibration at characteristic frequencies, chemical bonds of molecules can be utilized to obtain information of the molecules. Based on Raman active vibrational modes of molecules, coherent Raman scattering (CRS) is used for vibrational spectroscopic imaging method which has been an emerging technique to map the distribution of specific molecules inside specimen. The CRS has two types: coherent anti-Stokes Raman scattering (CARS) and stimulated Raman scattering (SRS), which has been used for important label-free imaging techniques in biology with high specificity of quantitative analysis.

### 2.1. Laser Scanning Coherent Anti-Stokes Raman Scattering Microscopy (CARSM)

In 1965, Terhune and Maker discovered the CARS phenomenon. They found a blue-shift of the Raman scattering spectrum when the frequency difference of two incident lasers match the benzene Raman scattering [10]. Then Duncan and coworkers demonstrated the first laser scanning CARSM in 1982 [11]. As shown in Figure 1a [12], the CARS is a coherent nonlinear optical process. Two laser beams (frequency $f_p$) with one for pump and probe and another for Stokes laser (frequency $f_s$), are impinged to target molecules within the sample. When the frequency difference between pump and Stokes laser satisfies the Raman active vibrational frequency of molecules $f_{vib} = f_p - f_s$, the CARS occurs with frequency being $f_{CARS} = 2f_p - f_s$. Specifically, the molecule absorbs a pump photon to a lower virtual state and emits a Stokes photon for returning to a higher energy ground state. Then the molecule immediately absorbs a probe photon to a higher virtual state, emits a photon with frequency of $f_p + f_{vib}$ and returns to the ground state, as shown in Figure 1a.

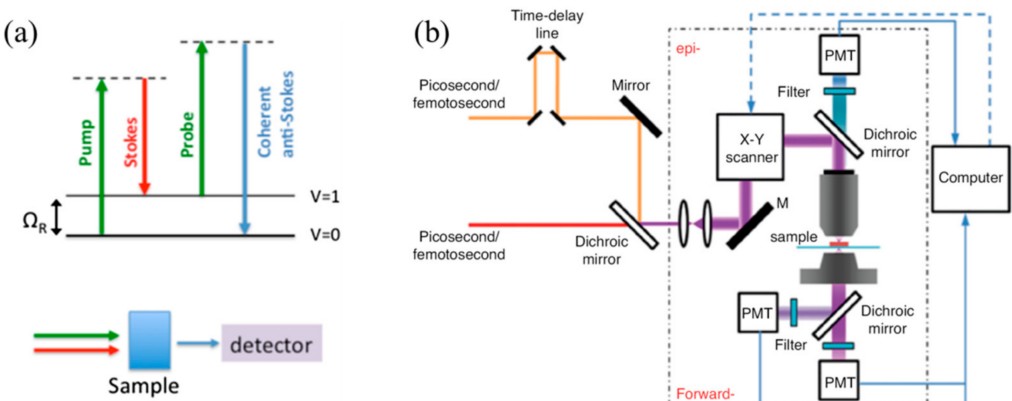

**Figure 1.** (**a**) Principle of coherent anti-Stokes Raman scattering (CARS) shown by Jablonski diagram (energy level diagram) [12] (reprinted with permission from [12], Copyright 2014 SPIE). (**b**) Diagram of a collinear laser-scanning CARS microscopy [13] (reprinted with permission from [13], Copyright 2018 SPIE).

The optical schematic of laser scanning CARSM is shown in Figure 1b [13]. Two incident lasers are collimated and tightly focused on sample to excite the CARS signal. A time-delay is used to control the temporal overlap of two excitation beams for effective excitation of CARS. The signal of backward propagating is collected by a high **NA** objective, which is also used to focus incident light. The signal of forward propagating is collected by another objective on transmission light path. The CARS signal is detected by PMT (Photomultiplier tubes) detector after filtering the incident lasers.

The CARSM began a rapid development via improving the detection speed, broadband, and sensitivity. Cheng et al. developed a high-speed CARS vibrational imaging method, which is two orders of magnitude faster than traditional CARSM by combining high repetition rate laser scanning and analog signal detection scheme [14]. Then a CARSM

with imaging speed of video-rate was developed for vibrational imaging of tissues in vivo by Xie's group via detecting strong backscattering of forward CARS signal with video-rate scanning microscopy [15]. Today, higher speed imaging has been achieved by employing more techniques, such as Fourier-transform [16] and deep-learning noise reduction [17]. The spectral bandwidth of CARSM has also shown great increase by multiplex [18,19] and broadband CARSM techniques [20,21]. Xie and coworkers reported a two-pulse multiplex CARS microspectroscopy using a picosecond pump beam and a femtosecond Stokes beam to demonstrate the polarization CARS imaging [18]. Lee et al. demonstrated the three-color CARS with continuum pulse of different frequency components as pump and Stokes and a narrowband pulse as probe [22]. As sensitivity of CARSM is restrained by the strong non-resonant background noise, polarization CARSM [23], phase and polarization coherent control CARSM [24], and femtosecond adaptive spectroscopic technique via CARS were introduced for suppressing background noise [25]. Cicerone's group developed a broadband CARS imaging platform by using intrapulse three-color excitation and exploiting the strong non-resonant background to amplify the inherently weak fingerprint signal, which provides an advantageous combination of speed, sensitivity and spectral bandwidth [26].

Recently, the study on improving the resolution of CARSM has attracted lots of attentions. By using laser beam shaping and higher-order CARS signal, resolution of CARSM has been remarkably enhanced. Resolution approaching 130 nm was achieved by narrowing the microscope's excitation volume in the focal plane through the combined use of a Toraldo-style pupil phase filter with the multiplicative nature of four-wave mixing [27]. A supercritical focusing CARSM has been developed by using two optimized phase patterns with combination of concentric rings with alternant 0 and $\pi$ phase, which is generated by a spatial light modulator and applied to the pump beam for minimizing the focal spot size [28]. The theoretical research has indicated that diffraction limit can be also broken by controlling the relative phase around the center of the pump and Stokes pulses [29]. Recently, a novel higher-order CARSM was investigated by using the higher-order nonlinear optical processes and detect the CARS signal at new wavelength to achieve high-contrast, super-resolution vibrational imaging [30].

Several distinct characteristics, including high specificity, high spatial resolution, high sensitivity, and label-free, make CARSM potentially ideal for noninvasively imaging complex biological and chemical samples. In living cells imaging, various molecule vibrational modes can be detected by CARS, such as phosphate stretch vibration, amide I vibration [14,31], OH stretching vibration [32,33] and CH stretch vibration [34,35], which can be applied for visualizing DNA, protein, water and lipids, respectively. Meanwhile, high imaging speed and signal level make CARS more suitable for observing biological dynamic process in vivo [15,36]. The synchronous change of the cell voltage and distribution of chemical states of water inside a nafion membrane was monitored with CARSM. For example, the transient values of the number of water molecules per sulfonic acid group increased rapidly from 7.4 at 0 s to 11.7 at 4.5 s under current density jump from 0.1 to 1.0 A·cm$^{-2}$ [37]. In addition, CARSM also found applications in gas phase analysis [38] and material characterization [39]. Multiplex CARSM imaging was demonstrated and used to visualize the thermodynamic state, including the liquid crystalline and gel phase of lipid membranes [19]. In addition, CARSM has also been a powerful characterization tool for Hexagonal Boron Nitride [40], porous carbon [41] and graphene [42].

### 2.2. Laser Scanning Stimulated Raman Scattering Microscopy (SRSM)

Although CARSM has good performance on high sensitivity and high spatial resolution imaging to biological and chemical specimen, it is not suitable for quantitative chemical imaging, and has to restrain the strong non-resonant background. As SRS signals depend linearly on concentration of signaling molecule inside sample with background-free contrast, SRSM is a good candidate for quantitatively analyzing the molecule map information inside specimen. The excitation process of SRS is displayed in Figure 2a [43]. Similar with CARS, when the frequency difference between pump and Stokes beams is equal to Raman

active vibrational frequency of molecules, amplification of the Raman signal is achieved by virtue of stimulated excitation. Consequently, the intensity of Stokes beam $I_s$, experiences a gain $\Delta I_s$ (stimulated Raman gain, SRG), and the intensity of pump beam $I_p$, experiences a loss $\Delta I_p$ (stimulated Raman loss, SRL), as shown in Figure 2b. Therefore, unlike CARS, SRL and SRG do not exhibit a nonresonant background. For SRSM, the Stokes beam is modulated at a high frequency, and the amplitude modulation of the pump beam can be detected by using SRL (Figure 2c). As shown in Figure 2d, the Stokes beam of SRSM is modulated by an electro-optic modulator. A photodiode is used to detect the transmitted and reflected light after filtering. SRL is measured by a lock-in amplifier to provide image pixels, and a fast scanning is performed for 3D imaging. Linear dependence of SRL signal intensity on concentrations of target molecule is shown in Figure 2e.

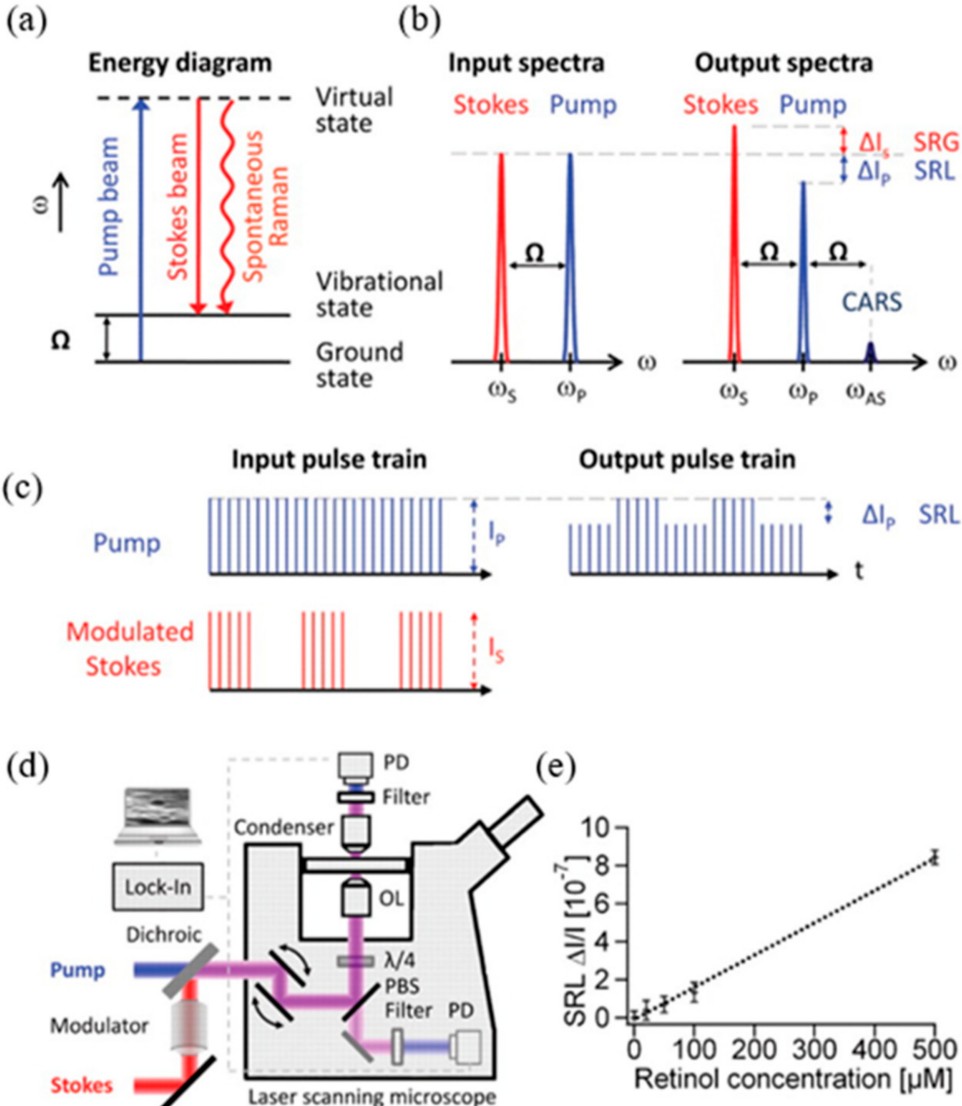

**Figure 2.** (**a**) Energy diagram for Stimulated Raman Scattering (SRS). (**b**) Input and output spectra of SRS. SRS leads to an intensity increase with a gain $\Delta I_s$ (stimulated Raman gain (SRG)) in the Stokes beam and an intensity decrease with a loss $\Delta I_p$ (stimulated Raman loss (SRL)) in the pump beam. (**c**) Detection scheme of SRL. (**d**) Stimulated Raman Scattering Microscopy (SRSM) with both forward and epi detection. (**e**) Linear dependence of SRL on concentrations of retinol in ethanol at 1595 cm$^{-1}$ [43] (reprinted with permission from [43], Copyright 2008 AAAS).

SRS was first discovered by Woodbury and Ng in 1962 [44,45]. Then SRSM was introduced by Ploetz et al. for imaging polystyrene beads in 2007 [46]. In 2008, Xie's group reported a high speed, high sensitivity, and three-dimensional multiphoton vibrational imaging technique based on SRS, which showed powerful label-free chemical contrast capability in bio-imaging [43]. The sensing range of typical SRSM is μM-mM of molecules concentration. Through using a microsecond resonant delay line, the detection sensitivity approaches 21.2 mM at 83 μs acquisition time [47]. Meanwhile, the super-resolution SRS imaging has been proposed and implemented by combining concepts from stimulated emission depletion (STED) microscopy and femtosecond stimulated Raman spectroscopy (FSRS) [48]. The far-field spatial resolution of SRSM has achieved ~130 nm with excitation wavelength of 800~1064 nm [49]. A virtual sinusoidal modulation method was also proposed for retrieving super-resolution SRS images, which demonstrated a spatial resolution of 255 nm [50]. As volumetric imaging provides global understanding of 3D complex systems, Chen et al. demonstrated the Bessel-beam-based simulated Raman projection microscopy and tomography for label-free volumetric imaging with spatial resolution being 0.83 μm [51]. Typically, the imaging depth of SRSM is 100 μm for high scattering biological samples and 300–500 μm for less scattering biological samples. Wei et al. developed a volumetric chemical imaging method that couples Raman-tailored tissue-clearing with SRSM, which achieves greater than 10 times depth increase compared with the standard SRS (Figure 3a) [52]. By now, SRSM has achieved a high speed with video-rate, sensitivity down to single molecules, spatial resolution breaking the diffraction limit and volumetric imaging depth with millimeter [53]. Moreover, SRSM permits rapid imaging of polystyrene and polymethyl methacrylate beads in Brownian motion in dimethyl sulfoxide (DMSO) at 70 ms intervals without motion artiacts, and shows powerful ability to monitor real-time cancer treatment effects and in vivo transport of drug solvent and penetration of DMSO into the plant tissue in vivo [54].

Recently, SRSM rapidly developed and was widely applied in various branch of life science, such as histopathology, cell biology, neuroscience, tumor research etc. Histopathological diagnosis in vivo has been demonstrated by using multicolor images originating from CH2 and CH3 vibrations of lipids and proteins, which retrieved the subcellular resolution from fresh tissue [55]. Moreover, SRS imaging to lipid metabolism was reported to reveal an aberrant accumulation of esterified cholesterol in lipid droplets of high-grade prostate cancer and metastases [56]. Label-free SRS imaging of DNA enabled noninvasive visualization of live cell nuclei in both human and animals. Through the distribution of DNA retrieved from the strong background of proteins and lipids by linear decomposition of SRS images at three optimally selected Raman shifts, it is possible to obtain instant histological tissue examination during surgical procedures [57]. SRSM has also been proved to be able to differentiate healthy human and mouse brain tissue as well as tumors from non-neoplastic tissue based on their different Raman spectra (Figure 3b) [58]. Likewise, SRS imaging has shown its potential capability and application for differentiating misfolded amyloid-β on Alzheimer's disease pathology (Figure 3c) [59]. These works provided new approaches for tracing dynamics and metabolism in living cells, which can be used for diagnosing and treating diseases.

SRS imaging has also been used for monitoring the breakdown of biomass to biofuels. It helps researchers to understand the conversion mechanisms of plant cell biomass to biofuels and find cost-effective industrial conversion strategy. By recognizing the Raman signal, the different chemical bonding is detected. A chemical movie recording degradation of lignin and cellulose in cell wall was used to understand which parts of plant are degraded most efficiently by the treatment process [60]. By comparing the two-colored SRS imaging before and after delignification reaction, another study revealed that hemicelluloses were generally resistant to acid chlorite at room temperature, and digestion of the delignified walls apparently depended on wall mass concentration but not on wall types (Figure 3d) [61]. These works have greatly promoted the development of SRSM and its application in biomedicine [62–65].

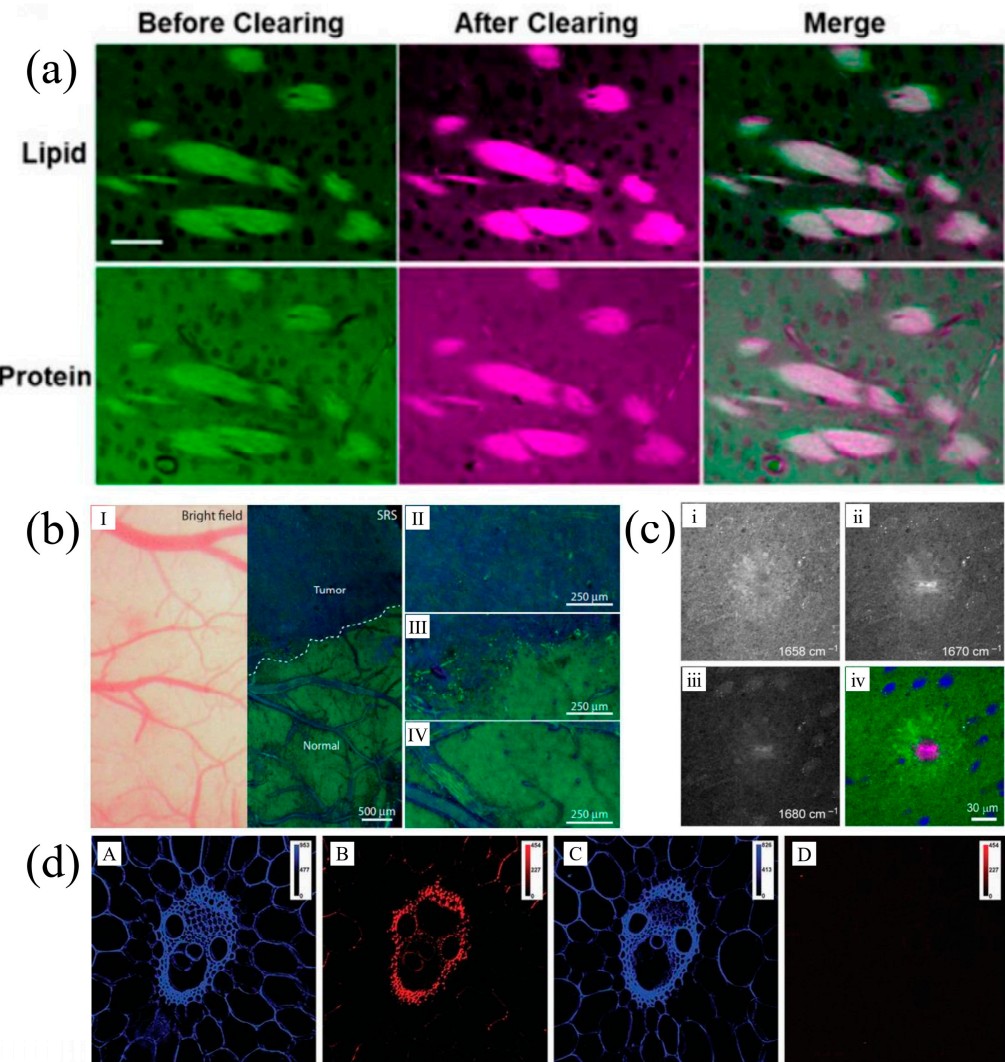

**Figure 3.** (**a**) SRS images of regions in the same white matter region before and after tissue-clearing [52] (reprinted with permission from [52], Copyright 2019 PNAS). (**b**) In Vivo SRSM images of human glioblastoma multiforme xenografts [58] (reprinted with permission from [58], Copyright 2013 AAAS). (**c**) Individual SRS images of a 1-mm-thick fresh mouse brain section at (**i**) 1658 cm$^{-1}$, (**ii**) 1670 cm$^{-1}$, and (**iii**) 1680 cm$^{-1}$ and (**iv**) the composite three-color image showing the distribution of lipid (green), normal protein (blue), and amyloid plaque (magenta) [59] (reprinted with permission from [59], Copyright 2019 AAAS). (**d**) SRSM of untreated cell walls [61] (reprinted with permission from [61], Copyright 2012 AAAS).

## 3. Second Harmonic Generation Microscopy (SHGM)

SHG process is a special case of the sum frequency process, also called "frequency doubling". As shown in Figure 4a [66], by using two incident lights with same frequency, the output is twice the frequency of the incident lights. The incident light is called fundamental frequency and the output with frequency doubling is called second harmonic. In 1961, SHG was first confirmed with a quartz sample by P. A. Franken et al. [67]. In 1974, Hellwarth and Christensen firstly integrated SHG and microscopy by imaging SHG signals from polycrystalline ZnSe [68]. A typical setup of SHGM is shown in Figure 4b [69]. The excitation wavelength was 780 nm. A galvo scanner was used to direct the beam in a raster-scan pattern. The beam was reflected by a dichroic mirror and focused onto the sample using a high *NA* water-immersion objective 1. The emitted backward SHG signal was collected by the same objective, whereas the forward signal was collected by a low *NA* objective 2. In both geometries the same two filters were used: one filter was used to block

the fundamental laser, and the other was a band-pass filter to transmit the SHG signal at 390 nm. The photomultiplier tubes in both geometries were used to record the forward and backward SHG images.

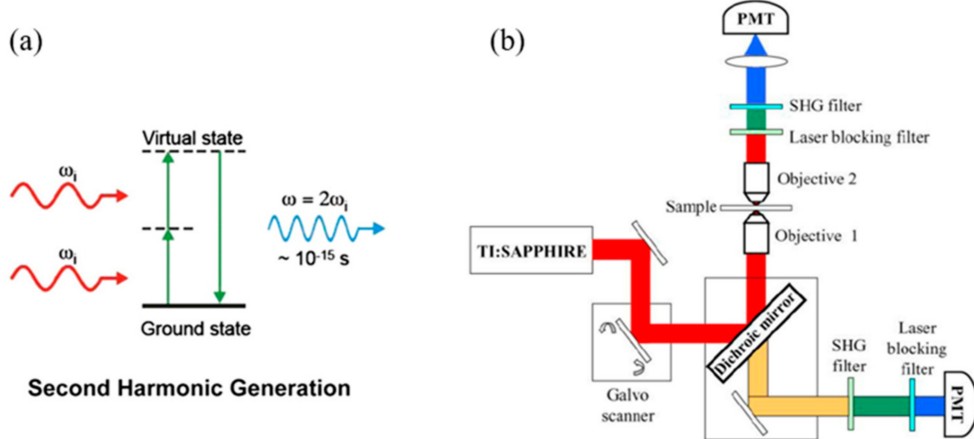

**Figure 4.** (**a**) Energy-level diagram of SHG [66] (reprinted with permission from [66], Copyright 2010 PNAS). (**b**) Experimental setup of SHGM with both forward and backward collection geometries [69] (reprinted with permission from [69], © The Optical Society).

In 1986, the first biological SHG imaging experiments were done by Freund and Deutsch to study the orientation of collagen fibers in rat tail tendon [70]. In 1993, Lewis examined the second-harmonic response of biological membrane in electric fields via using chiral styryl dye [71]. He also demonstrated his work on live cell imaging [72]. In 2010, P. Pantazis extended SHG to whole animal in vivo imaging [66]. The review by Paolo showed how SHGM achieved multimodal microscopic images of biological tissues, namely backward, forward and polarization SHG imaging [73]. However, the spatial resolution of SHGM is vastly restricted due to the near infrared excitation wavelength. Many methods have been carried out in order to improve the spatial resolution. Radially-polarized laser illumination is an effective and easy way for dramatically enhancing the resolution of SHGM [74–76]. Wang et al. demonstrated that spatial resolution was improved more than 21% by using radially-polarized beam focused by elliptical mirror compared with lens [77]. The work of Tian et al. showed that the subtractive imaging method could been extended to SHGM that increased the resolution to 210 nm and obtained a better contrast simultaneously [78]. Yeh et al. combined SHGM with structured illumination based on point-scanning, which improves the resolution by factors of ~1.4 in the lateral and 1.56 in the axial directions, respectively [74]. Sheppard proposed pixel reassignment, in which the point detector of the traditional confocal microscopy is replaced by a multi-element array detector [79]. Combining with deblurring operation, this method showed a spatial resolution enhancement of about 1.87 compared with conventional SHGM [80].

Nowadays, SHGM has been extensively used in visualization to structures and functions of cells and tissues. As SHG is highly sensitive to collagen fibril/fiber structure [70,81], SHG imaging can be used to detect collagen-associated changes occurred in various diseases [82], such as cancer, fibrosis, and connective tissue disorders. Dynamic reorganization of the collagen network in ex vivo murine skin dermis was imaged rapidly by polarization-resolved SHGM during mechanical assays with acquisition time of 27 ms per line [83]. Ovarian cancer was studied by SHGM to detect the structure changes of the ovarian extracellular matrix in human normal and malignant ex vivo biopsies (Figure 5a) [84]. The results showed that the SHG emission distribution and bulk optical parameters are related to the tissue structure and significantly different in the tumors. Combining with autofluorescence (AF), SHGM can prove a quantitative analysis to different collagen content in the process of tumor progression and recession post-chemotherapy via calculating

the ratio of SHG/AF (Figure 5b) [85]. Polarization-dependent SHGM was developed for accurately diagnosing the tumor state of breast cancer by analyzing anisotropy and the "ratio parameter" values of SHG signal (Figure 6a) [86]. Through further data analysis, parameters of PIPO (polarization-in, polarization-out) SHGM, i.e., susceptibility ratios and degree of linear polarization, can be used to distinguish tumor from normal tissue (Figure 6b) [87].

Furthermore, some automated or semi-automated solutions have been developed to analyze SHG images for evaluating the healthy and abnormal tissue. Early embryogenesis of zebrafish embryos was automatically reconstructed and analyzed with conformal scanning harmonic microscopy scheme for better imaging quality [88]. Fast Fourier transform analysis and the gray-level co-occurrence matrix have also been applied to analyze biopsy samples of human ovarian epithelial cancer at different stages [89]. It showed the feasibility of identifying collagen fibril morphology based on first and second-order texture statistical parameters from SHG images. In addition, radiomics feature analysis, tree-based pipeline optimization tool [90] and 3D texture analysis based on k-nearest neighbor classification are also proposed for differentiating healthy and pathologic tissues [91].

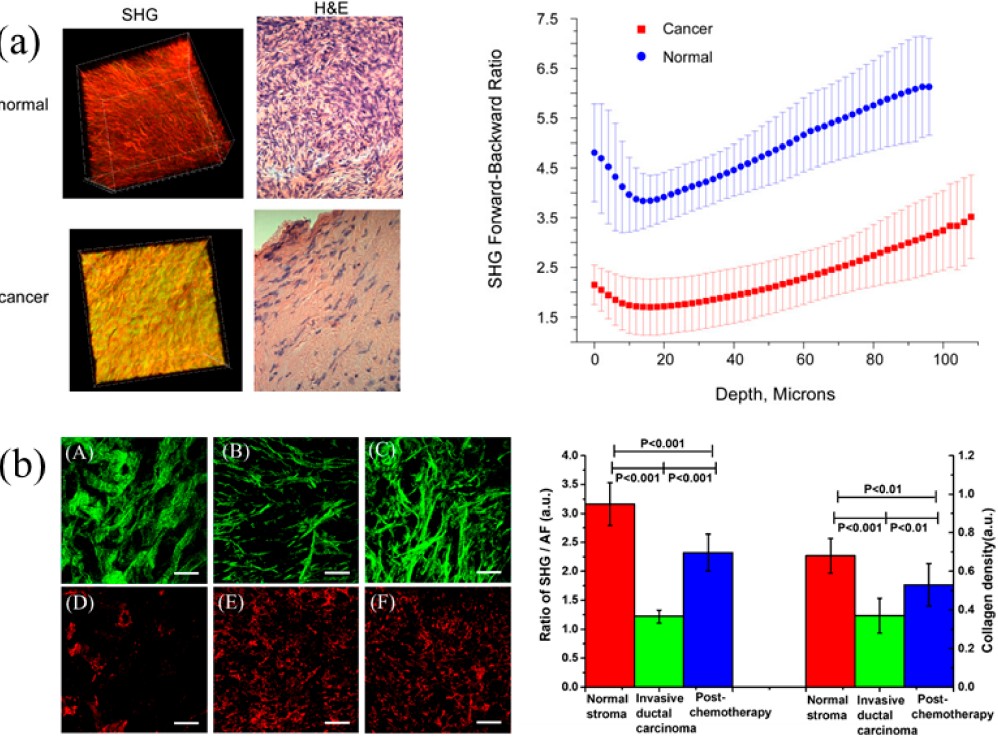

**Figure 5.** (**a**) 3D Second Harmonic Generation (SHG) renderings (left panels) and hematoxylin and eosin (H&E) staining (center panels) of normal (top) and malignant ovarian biopsies (bottom). Averaged forward/backward SHG intensities as a function of depth for normal (blue circles) and malignant (red squares) ovaries (right panels) [84] (reprinted with permission from [84], Copyright 2010 Springer Nature). (**b**) SHG images and correcting autofluorescence (AF) images of the collagen in different tissues status (left panels). SHG images of (A) normal stroma, (B) invasive ductal carcinoma, (C) post-chemotherapy. AF images of (D) normal stroma, (E) invasive ductal carcinoma, (F) post-chemotherapy. The ratio of SHG/AF and the collagen density of the three status of breast tissues (right panels) [85] (reprinted with permission from [85], © The Optical Society).

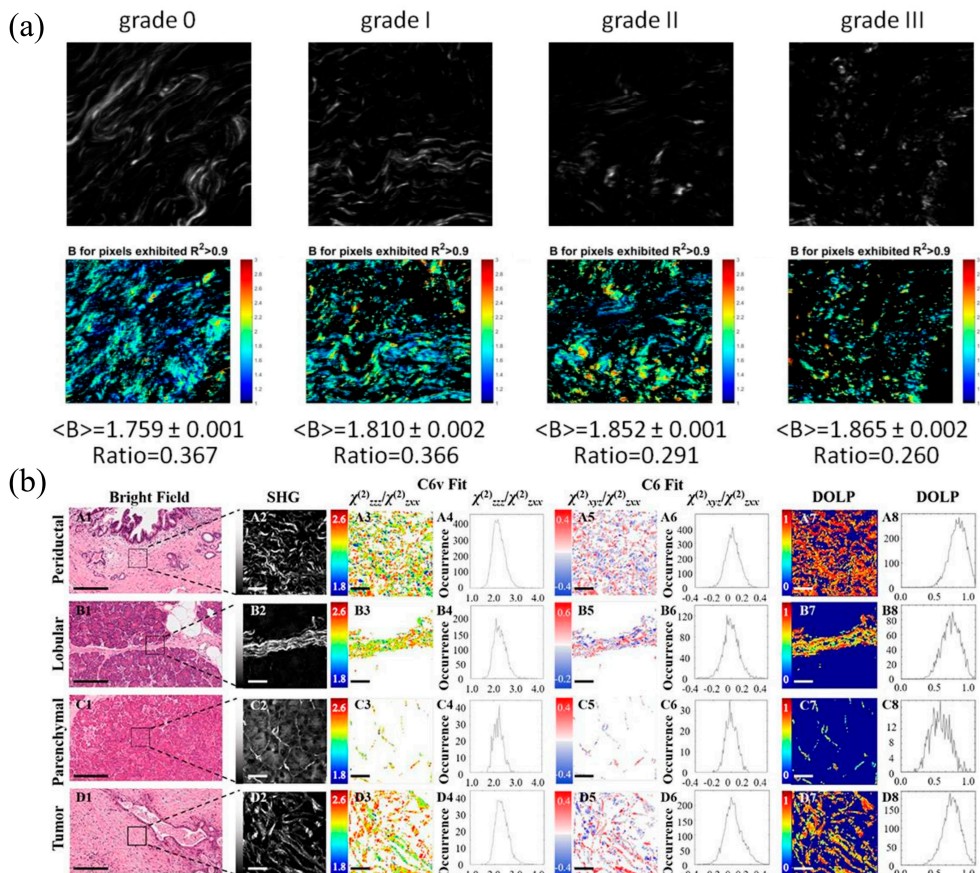

**Figure 6.** (**a**) Representative examples of FFT analysis for benign (grade 0), grade I, grade II and grade III breast tissues [86] (reprinted with permission from [86], Copyright 2020 Wiley-VCH). (**b**) PIPO (polarization-in, polarization-out) SHGM of collagen in (A1–A8) Periductal, (B1–B8) Lobular, (C1–C8), Parenchymal in pancreas tissues, and (D1–D8) Tumor with (A1, B1, C1, D1) Bright-field images, (A2, B2, C2, D2) SHG intensity images, (A3, B3, C3, D3) Color-coded maps of the fitted R values, (A4, B4, C4, D4) Occurrence histograms of the R values, (A5, B5, C5, D5) Color-coded maps of the fitted C values, (A6, B6, C6, D6) Occurrence histograms, (A7, B7, C7, D7) Degree of linear polarization values, and (A8, B8, C8, D8) Occurrence histograms [87] (reprinted with permission from [87], Copyright 2019 Frontiers Media SA).

## 4. Scanning Localized Surface Plasmon Microscopy (SLSPM)

Surface plasmon polaritons (SPPs) is the collected oscillation of electrons at metal-dielectric interface. As SPPs is evanescent wave along metal-dielectric interface, the excitation of SPPs requires the wavevector matching condition satisfied by prism, waveguide, and gratings. Due to their high sensitivity to refractive index change near interface, in 1984, SPPs was firstly used as gas detector and bio-sensor based on Kretschmann configuration [92,93], then SPPs became an essential tool for biological and chemical sensing [94–97]. Due to non-intrusive, label-free and high sensitivity surface plasmon microscopy (SPM) is a suitable technology for near-field imaging to the cell-substrate interface [98]. However, the resolution of SPM is limited by propagation length of SPPs, e.g., being 7 μm for gold film illuminated by 633 nm wavelength. In order to improve the resolution, H. Kono et al. firstly presented SLSPM, in which the SPPs is excited by a focused laser beam as sensing probe to measure refractive index and film thickness [99,100]. Polarization of incident light is an important factor to increase resolution of SLSPM. Axially symmetric radial-polarized light optimizes the probe with full width at half maximum of 210 nm that increases the resolution [101]. H. Kono et al. used the SLSPM to image microspheres with diameter being 1.5 μm [100]. Nowadays, SLSPM have improved lateral resolution of SPP imaging up to ~200 nm by using high numerical aperture (*NA*) objective [96,101].

The setup of SLSPM is shown in Figure 7a [102], the collimated incident light illuminates objective after transforming through lenses, mirrors and radial polarizer, which ensures that incident light is TM-polarization in each incident plane of focal beam. The incident light is focused into spot with diffraction limit dimension to excite SPPs. The maxima of incident angle is determined by **NA** of objective as $NA = n\ sin\theta_M$ ($\theta_M$ is maximum angle of incident light). The intensity distribution on back focal plane (BFP) is recorded by a CCD camera, and a high-resolution image of specimen is acquired by using two-dimensional scanning. Figure 7b shows how the SPP evanescent field penetrates the cytosol on gold film. A typical BFP image is shown in Figure 7c, in which the dark ring indicates the SPPs excited by the incident light and ring radius represents propagation constant of SPPs. Along the radial direction, the reflective intensity profile dependent to incident angle is illustrated in Figure 7d. The refractive index and thickness of specimen can be obtained by fitting the resonant angle using a theoretical multilayer model [103,104].

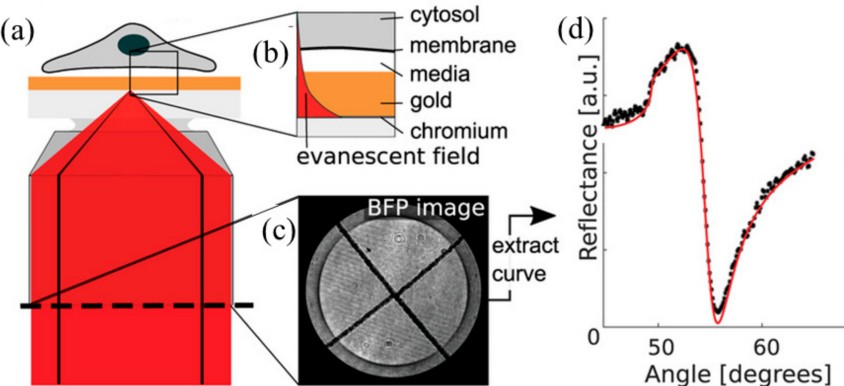

**Figure 7.** (**a**) The entire back focal plane (BFP) is illuminated with radially polarized light resulting in an illumination of the sample with a broad range of angles. (**b**) Illuminating the sample under surface plasmon polaritons (SPP) excitation angles induces the SPP evanescent field penetrating the cytosol. (**c**) The BFP imaging at each scanning point allows for the extraction of (**d**) reflectance curve [102] (reprinted with permission from [102], Copyright 2020 Wiley-VCH).

To gain better imaging quality and greater application, researchers have made many attempts to improve the performance and expand its application range. SLSPM was demonstrated to detect oligonucleotide-functionalized nanoarray by S. C. Wei, et al. [105]. In order to improve the image contrast, M. G. Somekh et al. introduced a Z-direction movement mechanism to the two-dimensional scanning mechanism. They found that if the sample is slightly shifted from the focus by about 1 μm, the image contrast (refractive index sensitivity) will be improved and the refractive index can be retrieved from the interference fringes [106,107]. In 2012, B. Zhang et al. proposed a confocal system integrated scanning SPM, which provides a simpler and more stable alternative [108]. L. Berguiga et al. reported a high-resolution metal-clad waveguide scanning microscopy with a diffraction-limited resolution, which can be operated in both TM and TE waveguide modes with radially and azimuthally polarized beams, respectively, and allows both refractive index and topography of dielectric objects to be evaluated at high resolution and sensitivity [106]. K. Watanabe reported a high resolution imaging with lateral resolution of ~170 nm and thickness resolution against the deposited lipid bilayer being ~0.33 nm [109]. E. Kreysing et al. proposed a workflow to make SLSPM as a complement for EM studies (Figure 8a, which allows for the quantification of the impact of chemical fixation on the cell–substrate interface [102].

As SPPs is evanescent wave, the SLSPM is particularly attractive for studying cell adhesion and dynamics without staining the samples [110–112]. Time-lapse long-term imaging of living adherent cells was reported via combining SLSPM and fiber interferometer [113]. Adherence and motility of living C2C12 myoblast cells are monitored for

50 h, revealing the dynamics of these cells during their migration (Figure 8b) [114]. Further understanding at the bio-interface has growing importance in fundamental neuroscience and biomedical research. With SLSPM, cell-substrate interface was real-time monitored and its cleft gap distance was measured in-situ [115]. E. Kreysing et al. showed a temporal and spatial mapping to changes of both refractive index and the cell-substrate distance which strongly correlates with the action potentials. They also reconstructed the three-dimensional profiles of the basal cell membrane and its dynamics, which reached an actual measurement accuracy of 2.3 nm (Figure 8c) [103]. By using SLSPM, they also analyzed the refractive index of cell organelles quantitatively in a noninvasive and label-free manner with a lateral resolution of diffraction limit [104].

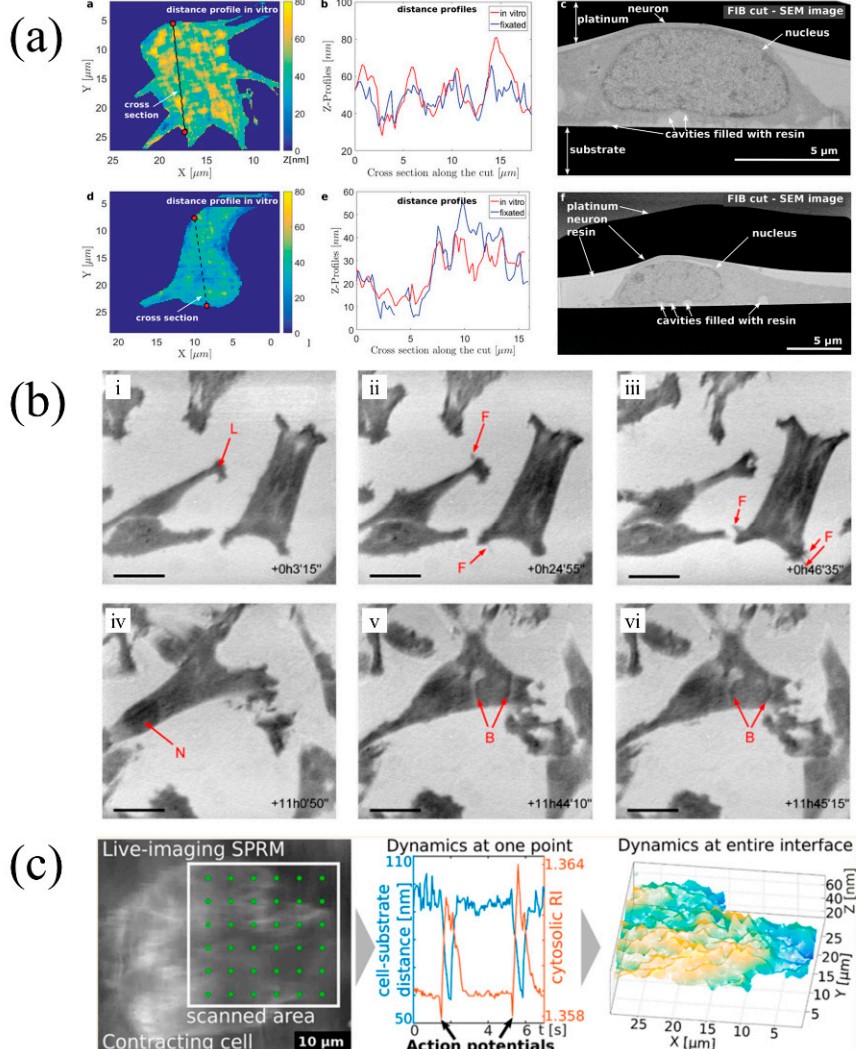

**Figure 8.** (**a**) a, d, Correlation of scanning localized surface plasmon microscopy (SLSPM) and electron microscope (EM) for two individual cells. b, e, The cell substrate distances are evaluated along the cross-section with surface plasmon resonance microscopy (SPRM). c, f, Focused ion beam milling combined with scanning electron microscopy (FIB-SEM) for the same cells, [102] (reprinted with permission from [102], Copyright 2020 Wiley-VCH). (**b**) (i)–(vi) SLSPM images of living C2C12 cells at different times. The arrows have the following meaning: filopodia (F), lamellipodia (L), nucleus (N) and fronts (B) [114] (reprinted with permission from [114], © The Optical Society). (**c**) Nanometer-resolved mapping of cell-substrate distances of contracting Cardiomyocytes. Cell image under live-imaging mode and scanned area (white rectangular box) (left), cell−substrate distance and refractive index change over time (middle), morphology distribution under SLSPM [103] (reprinted with permission from [103], Copyright 2018 ACS Publications).

## 5. Discussion

In this review, three types of label-free laser scanning microscopy are mentioned with their particular features. The laser scanning CRSM is a vibrational spectroscopic imaging method by using Raman active vibrational modes of molecules, which has two subtypes: CARSM and SRSM. With second-harmonic response of biological samples, SHGM is achieved and has highly sensitivity to the collagen fibril structure, which can be used to detect collagen-associated changes occurred in various diseases. Due to SPPs' evanescent features, SLSPM is highly sensitive to refractive index change near the interface, which is appropriate for investigating cell-substrate interface. Label-free laser scanning microscopies show powerful capability for imaging and dynamic tracing biological tissues and cells. Imaging speed and sensitivity are key issues for the label-free LSM methods. Moreover, spatial resolution is also an important parameter for the microscopies. We organized the comparison of three label-free LSMs, which is shown in Table 1. The lateral resolutions of the three label-free LSMs possess the same magnitude, while the axial resolution of SLSPM is extremely high which is below one nanometer.

**Table 1.** Comparison of three laser scanning microscopic technologies.

| Type | | Work Principle | Work Wavelength (nm) | Repetition Rate | Laser Power | Lateral Resolution (nm) | Axial Resolution | Ref. |
|---|---|---|---|---|---|---|---|---|
| **CRS** | CARSM | supercritical focusing CARS | 835 (pump) 1107 (Stokes) | 76 MHz | 3 mW (pump) 10 mW (Stokes) | 295 | 3.32 μm | [28] |
| | | coherent controlling of the relative phase | 785 (Stokes) 727.8 (pump) | - | - | 130 | - | [29] |
| | | higher-order CARS | 1041 (Stokes) 680–1300 (pump) | 80 MHz | $2.0 \times 10^9$–$3.6 \times 10^{12}$ W/cm$^2$ (pump) $1.8 \times 10^9$–$4.7 \times 10^{12}$ W/cm$^2$ (Stokes) | 196 | - | [30] |
| | SRSM | near-resonance enhancement | 450 | 80 MHz | 4.2 mW (pump) 2.6 mW (Stokes) (brain tissue) | 130 | - | [49] |
| | | Saturated SRS | - | - | $227 \times 10^9$ W/cm$^2$ (pump) $214 \times 10^9$ W/cm$^2$ (Stokes) | 255 | 3 μm | [50] |
| **SHGM** | | structured illumination | 1064 | 40 MHz | 30 mW | 235 | 693 nm | [74] |
| | | subtractive imaging | 800 | 80 MHz | - | 210 | - | [78] |
| | | pixel reassignment | 840 | - | - | 300 | - | [80] |
| **SLSPM** | | scanning localized SPP | 632.8 | - | - | 186 | 2.3 nm | [103] |
| | | scanning localized SPP | 632.8 | - | - | 170 | 0.33 nm | [109] |

Although label-free LSMs prevent the photodamage and photobleaching induced by high intensity laser and fluorescent molecule, photodamage and photobleaching are also key issues that should be considered. In fluorescent confocal laser scanning microscopy (FCLSM), single-photon absorption and heating are the dominant sources of photo-induced damage, which carries the risk of linear damage [116]. The power density of $3.0 \times 10^8$ W/cm$^2$ induced the photodamage to zebrafish craniofacial bone by using FCLSM [117]. For label-free LSMs, namely CARSM, SRSM and SHGM, the nonlinear effects require illumination with high power, which might also lead to photodamage and photobleaching. Some strategies have been employed to minimizing the photodamage and photobleaching in the nonlinear label-free LSMs. For CARSM, they are negligible by using excitation light with long wavelength, proper power and repetition rate [118–120]. In SRSM, nonlinear absorption is avoided for reducing the risk of photodamage and photobleaching because of background-free and long wavelength for excitation [118,121]. Furthermore, as

involving only virtual state without energy deposition, SHGM brings little photodamage or photobleaching by increasing the repetition rate that keeps the peak intensity below damage threshold [122,123]. The laser powers and repetition rates of label-free LSMs are indicated in Table 1.

However, there are still some limitations for current label-free laser scanning microscopies. CARSM is relatively easy to implement but with non-resonance background from a four-wave mixing process inside sample. This impedes accessibility of CARSM to higher resolution and sensitivity in biological imaging. Compared to CARSM, SRSM is background-free imaging, but the extraction of SRS signal requires complex lock-in detection for modulation and demodulation. SHGM is suitable for samples without inversion symmetry, such as collagen and myosin, which limits its scope of application. SLSPM is limited to image samples within evanescent penetration depth of SPPs and have a slow imaging speed due to scanning mode and sophisticated data process.

From the above discussion, no single imaging technology can present a global understanding to the complex biological structure and process. Therefore, multi-technique combination is a trend for imaging and information analysis to complicated samples [124–126]. New imaging schemes will make these label-free laser scanning microscopies to be more powerful biological imaging tool when combine with spectroscopy, laser and detection technology. Higher resolution, sensitivity and faster imaging speed are still the most important pursuits of microscopy for more detailed and accurate morphology and dynamic process. With the increase of imaging speed and broadband, more effective and automated data method is required to process statistics and extremely large amounts of raw data, and will benefit tremendously from growth of artificial intelligence; fast and automated data processing will become easier to implement. There is a reasonable prospect that advanced label-free laser scanning microscopy will occupy a more dominant position in biological and biomedical imaging.

**Author Contributions:** Conceptualization, X.W., X.L.; writing—original draft preparation, X.W., X.L.; writing—review and editing, X.W., X.L., C.H.; supervision, C.H. All authors have read and agreed to the published version of the manuscript.

**Funding:** This research was funded by National Key Research and Development Program of China, grant numbers 2018YFC2001100 and 2017YFF0107002; Beijing Natural Science Foundation, grant numbers 4192063 and 4182073; Scientific Research Equipment Project of Chinese Academy of Sciences, grant numbers YJKYYQ20190056.

**Institutional Review Board Statement:** Not applicable.

**Informed Consent Statement:** Not applicable.

**Data Availability Statement:** Data sharing not applicable.

**Conflicts of Interest:** The authors declare no conflict of interest.

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
