# Peer review of "Advanced Label-Free Laser Scanning Microscopy and Its Biological Imaging Application"

_applsci, doi:10.3390/app11031002_

Round 1

Reviewer 1 Report

The work represents a good and deep review of the main label-free scanning microscopy methods that offers the possibility to overcome the diffraction limit. 

In general, the work is well written and the literature analyzed and reported is quite extended.

In section 3, I believe that a reference to an important work is missing (Bianchini, P., and A. Diaspro. 2008. Three-dimensional (3D) backward and forward second harmonic generation (SHG) microscopy of biological tissues. J. Biophotonics. 1:443–450.9).

I would recommend for publication after minor changes to the English language, in particular:

  • line 56 technique
  • line 90 shows
  • line 248 
  • line 362 of-> at the
  • line 365 molecule, shows
  • line 366 to
  • line 367 type
  • line 386 limit
  • line 393 combines

Reviewer 2 Report

The article by Xue Wang is devoted to a review and comparison of three promising methods of probeless scanning optical microscopy used in biology: laser scanning coherent Raman microscopy, scanning microscopy with second harmonic generation, and scanning surface microscopy of plasmon resonance. The article discusses the differences between the presented methods and the methods of confocal fluorescence scanning microscopy that are familiar to experimental cell biologists. The authors describe in detail the physical foundations of these methods and the methodological features of constructing experimental installations. A separate section is devoted to examples of the application of these methods for solving various biological problems, including cancer diagnostics and cell adhesion. The disadvantages of work include the following points:

1) the authors concentrate on the analysis of samples stable in time, without discussing at all the possibilities of these methods for the analysis of dynamic parameters, for example, changes in the concentration of biomolecules in cells in the course of a response to a stimulus of physical or chemical nature (this needs to be corrected);

2) in the abstract of the article, as an advantage of the methods presented, the absence of photodamage of the samples under study is given, while nowhere in the article is an assessment of the possibility of such damage, data on the flux density of electromagnetic energy of light in the corresponding study are given, there is no comparison of this characteristic with classical fluorescence microscopy. This is important, especially considering the fact that the radiation powers used in these experiments (especially when generating the second harmonic) lead to nonlinear effects during light scattering. It would be appropriate to place these data in Table 1. In addition, it is possible to discuss photodamage when using the presented methods on the basis of literature data, for example:

Marcos A. S. de Oliveira, Zachary J. Smith, Florian Knorr2, Renato E. de Araujo, and Sebastian Wachsmann-Hogiu. Long term Raman spectral study of power-dependent photodamage in red blood cells. Appl. Phys. Lett. 104, 103702 (2014); https://doi.org/10.1063/1.4868253

Yan Fu, Haifeng Wang, Riyi Shi, and Ji-Xin Cheng. Characterization of photodamage in coherent anti-Stokes Raman scattering microscopy Optics Express Vol. 14, Issue 9, pp. 3942-3951 (2006) • https: //doi.org/10.1364/OE.14.003942

In general, the article is useful and can be recommended for publication when eliminating these shortcomings.
